# A Decision-Making Model on the Impact of Vehicle Use on Urban Safety

**Dariusz Masłowski \*** , **Małgorzata Dendera-Gruszka and Ewa Kulińska**

Faculty of Production Engineering and Logistics, Opole University of Technology, 45-267 Opole, Poland;
m.dendera-gruszka@po.edu.pl (M.D.-G.); e.kulinska@po.edu.pl (E.K.)
\* Correspondence: d.maslowski@po.edu.pl

**Abstract:** In the current era of urban development, people are already using electric vehicles more and more often for transport purposes, which reduces negative impacts on the environment. However, there are still vehicles in towns and cities that run on ordinary internal combustion engines. Performing optimization measures on the operation of these vehicles improves their performance, which can result in positive sustainable development effects. This article presents measures to reduce the wear and tear of urban vehicles and outlines a decision model to determine which of the vehicle parts described suffer the most frequent wear and tear under urban conditions. The article presents a list of structural elements that are most affected by urban traffic, as well as corrective actions to improve such specialized vehicles. Based on the decision analysis, Rule 1 was eliminated as having the least significant impact on vehicle wear and tear, and the least significant impact on urban safety. On the other hand, the most worn-out elements were found to be gearboxes, clutches, bus levelling electronics, and brake pads and discs. The decision-making model made it possible to identify the factors which have the greatest impact on reducing safety in urban spaces.

**Keywords:** sustainable logistics; urban buses; bus operation; restrictive measures; decision-making model; urban safety





## 1. Introduction

Nowadays, urban travel is becoming a major challenge. The continuous shift of the population to cities and, consequently, the increased number of urban motor vehicles causes great difficulties in the form of traffic congestion. Congested cities require drivers to be highly skilled in driving in such a way as to avoid collisions or road accidents. Due to such a specialized driving style being needed, the least advantageous element of the urban driving journey is the increased wear and tear of vehicle components [1]. The most affected group of vehicles is city buses, which travel hundreds of kilometres around their cities daily (depending on the area of those cities), during which time the buses often struggle with the difficulties of traveling through such large centres. The example of city buses is used to underpin an analysis of vehicle wear and tear in this article. Almost every element of these vehicles is subject to wear and tear, beginning with driving wheels, through the braking system, to the steering system. The whole process of wear and tear is most often referred to as the process of gradual destruction of parts under the influence of physicochemical factors operating for the whole period of use of the parts [2–5]. The wear and tear of motor vehicle parts also occurs in normal, otherwise proper operation, and it is an unavoidable phenomenon, though with a higher intensity of external factors, this process accelerates [5,6].

Decision making in logistics is an increasingly complex task for organizations, as it involves decisions at strategic, tactical, and operational levels coupled with the triple-bottom line of sustainability. Decision support systems arguably play a vital role in solving the challenges associated with decision making in sustainable logistics [7,8]. The basic

assumption is that in order to make logistics sustainable, most of it needs to be rediscovered [9]. Logistics works well in many areas, regardless of the tools that are used. Sustainable development has been defined by the World Commission on Environment and Development (WCED) as development that meets the needs of the present without compromising the ability of future generations to meet their own needs [10]. Therefore, sustainable logistics can also be defined as a more efficient use of available logistics solutions and the introduction of optimization measures in this area [11].

The experience of the world's major cities shows that sustainable public transport is the most effective public transport solution in the long term [12]. Action is needed to reduce negative urban impacts of public transport. These actions are understood as the use of technological solutions as well as decision making solutions in relation to the implementation of appropriate tools. In the era of urban development, transport already uses electric or solar vehicles more and more often [13], which reduces the negative impact on the environment. However, there are still buses in towns and cities that operate with ordinary internal combustion engines. Performing optimization measures on their operation improves their performance, which can result in sustainable development effects [14]. If vehicles are unable to operate in an urban environment, this can result in a variety of additional costs having a negative impact on cities [15].

This article presents measures to reduce the wear and tear of urban vehicles and the development of a decision model to determine which of the vehicle parts described suffer wear and tear most frequently in urban conditions. The article presents a list of structural components that are most affected by urban traffic, as well as corrective actions for driving such specialized vehicles.

## 2. Research Methods

Decision boards are used to document and analyse systems, which are complementary to the classic schemes of modification and solution search. The basis for the construction of decision boards is the condition: "If ..., then ...", which is why they enable the construction of automatic information processing diagrams. Decision boards consist of a set of rules that are supposed to describe what set of conditions must be met in order for the proper scope of activities to be undertaken [16,17].

In particular, the so-called excess of conditions (decision ambiguity for the same actions) and contradiction of conditions (decision ambiguity for the same conditions) should be excluded. In practical applications, it is convenient to do a decomposition of a decision table into a dendrite with the assumption that the obtained tables only have one decision rule, regardless of the number of conditions in them. Due to the formalization of such a procedure, various logical additional definitions are used: Column numbers, dash numbers, delta parameters, common paths, common path powers, etc.

The generation of new solutions by the method of decision tables can be coded into binary and multivalued forms, which enables the subsequent correct circulation of information for true solutions, true sub-solutions, and most importantly true sub-solutions. Table 1 presents a standard decision table.

The set of activities defines the expressions "If...", which define the variables that have the greatest impact on the decision-making process. The set of actions describes the expressions "then...", which contain all the possible actions. The set of conditionality indicators is described by the symbols Y (Yes), N (No), and '-' (nothing), which specify whether or not a particular condition is met. A set of activity indicators is described in the same way as a set of conditionality indicators. On the other hand, the factors are described by the symbols X and '-'.

**Table 1.** Example of a decision board [16].

| Name of the Table | | Rules of Decision |
|---|---|---|
| | | $R_1, R_2, R_3, \ldots R_n$ |
| Set of conditions | $W_1$ $W_2$ $W_3$ $\ldots$ $W_n$ | Set of conditionality indicators |
| Set of activities | $C_1$ $C_2$ $C_3$ $\ldots$ $C_n$ | Set of activity indicators |

## 3. Wear and Tear on Urban Vehicle Components

There are many elements that wear out while driving city vehicles. However, the extent of wear and tear of the parts depends primarily on the service life as well as the mileage of the vehicle (intensity of use). The intensity of wear in periods of time can be illustrated by the Lorentz curve (Figure 1), which may lead to a conclusion that the process of wear and tear relative to time is not the same [5].

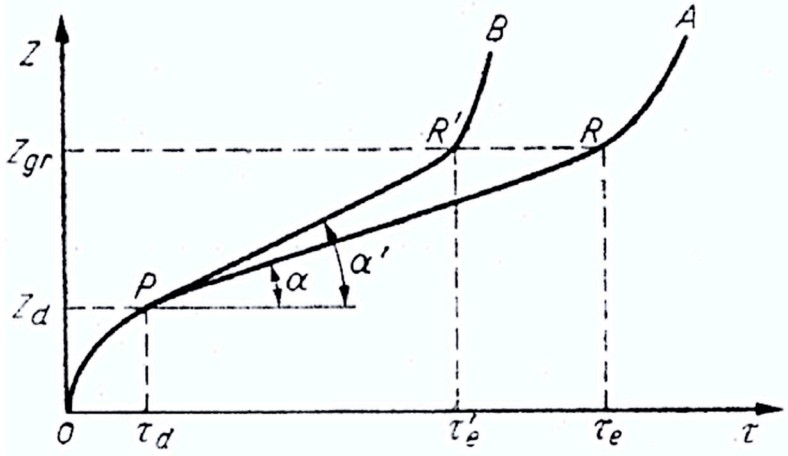

**Figure 1.** Curve of wear and tear of a bus part [2].

The axis of ordinates shows the amount of Z parts wear, and the cut-off axis—the operating time (operating course) $\tau$. For the sake of simplicity, only one part has been analysed for wear and tear. Wear of other parts of the relationship, e.g., the kinematic node or the mating pair, can be similar in nature. Z wear can be measured in different units (e.g., linear, mass, volumetric wear, play gain, etc.) However, for a certain number of vehicles operated under different conditions and with proper maintenance, the wear pattern of parts according to the OPR'B curve may be treated as normal wear for the given operating conditions [2,19].

Taking into account the curve presented, the individual sections differ in the extent of wear and tear. The causes of accelerated wear may include [5]:

- different operating conditions:

    ○ Road conditions,
    ○ field conditions,
    ○ climatic conditions,
    ○ the kind of work you do,

- improper maintenance:

&#9675; Using the wrong lubricants,
&#9675; using the wrong oils,
&#9675; incorrect performance of technical inspections,
&#9675; underestimating the initial signs of wear, and
&#9675; failure to comply with the manufacturer's recommendations.

Figure 2 shows a diagram of a vehicle with a list of elements that are most vulnerable to wear during urban driving [20,21].

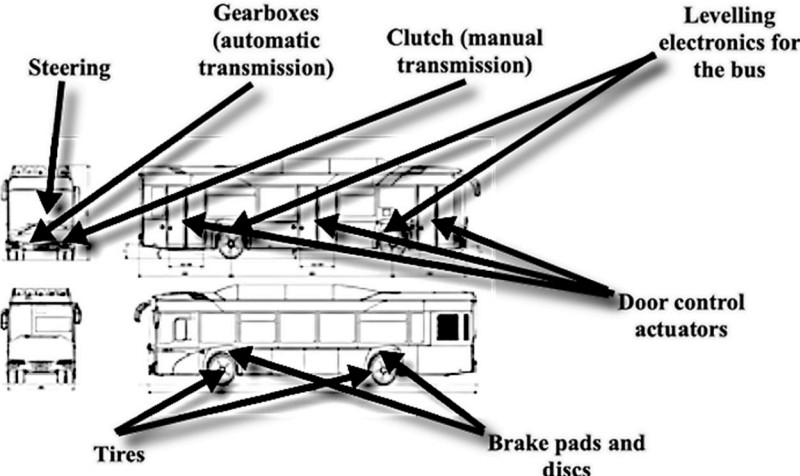

**Figure 2.** Bus diagram with a list of the most vulnerable components [Own study].

The article will analyse the above-mentioned parts used in vehicles moving around in an urban space, with particular emphasis on the proposed measures to reduce the process of wear and tear of elements.

### 3.1. Bus Tyres

Tyres are one of the most important parts in city vehicles, especially in vans and buses (Figure 3), as it is here that all the forces are transferred between the road and the means of transport, and they also suppress unevenness of the ground [3].

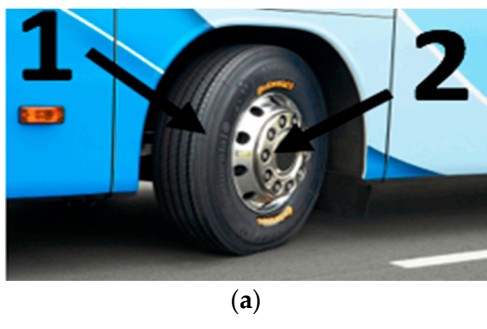
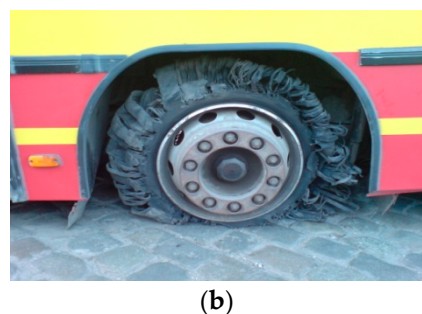

(**a**)                  (**b**)

**Figure 3.** (**a**) Construction of the wheel 1—tyre, 2—rim, (**b**) a damaged bus tyre [22].

The wear of this element is therefore an important factor that can affect values such as:

- Fuel consumption while driving,
- wear and tear of vehicle suspension components,
- comfort of travelling,
- convenience of driving,
- the number of accidents caused by a tyre failure,
- number of vehicle inspections.

The process of construction and selection of tyres is very complicated and important for the safety of vehicle users, as the tyres at the back of vehicles wear out one and a half times faster than the tyres on the axles at the front of the vehicle [23]. The tyres used in urban traffic contribute to the intensification of tyre wear, where the following frequent processes are performed [24]:

- Braking and accelerating,
- turning the wheels while manoeuvring,
- rubbing the sides of the tyre against the kerbs,
- implementing courses of fully fledged vehicles.

In addition, hitting the kerbs has a negative impact on the tyre structure and shortens its service life despite a sufficient tread depth [25].

In order to extend the life of the tyres, drivers and mechanics should pay particular attention to the obligatory special control of tyres as part of daily service, regardless of the level of automation of service and control processes in vehicles [23].

### 3.2. Braking System

The braking system is a key system in a car that ensures that vehicles function properly on the road. Brake discs are a very important part of the braking systems of all the commercial vehicles. There are brake pads that are pressed against the brake discs when the vehicle driver initiates the braking process. The result is friction, and the wheels reduce their speed. What makes the discs different from the rest of the parts is that they are movable, i.e., they spin at the same speed as the wheels and axles. Due to friction, the brake discs have to be checked and replaced on a regular basis as they wear out and may not perform their work as intended after a certain period of time. This is why when choosing brake discs for vehicles, it is particularly important that they are manufactured by a renowned company which uses only the best materials [26].

A braking system is composed of all the components and systems in a vehicle which are intended to stop it. It normally consists of the following parts (Figure 4):

- Front brakes (brake pads and discs);
- rear brakes (drum and brake disc);
- central part (pump and brake pedal);
- and the cables connecting all the components of the system to each other.

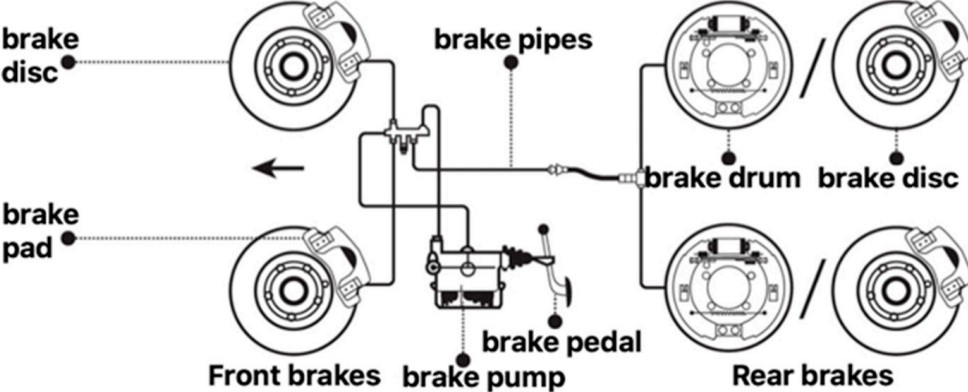

**Figure 4.** Simplified construction diagram of the drum/disc brake system [own study based on [27]].

In the automotive industry, the following braking system components can be distinguished:

- Front brakes (brake pads and discs)
  - the brake mechanisms:
  - Shoe and drums,
  - tape,
  - disc,

- actuators:
  - Mechanical,
  - hydraulic,
  - pneumatic vacuum,
  - pneumatic overpressure,
  - electropneumatic.

In vehicles, however, the most commonly used type of actuation systems are positive pressure systems, due to the ease of obtaining the considerable forces necessary to actuate the braking mechanisms of the wheels of such vehicles. Owing to the considerable height of the working pressure, usually 49–88 Pa, high braking forces are achieved with small dimensions of brake actuators and other components of the system.

In practice, two systems offering such solutions can be distinguished:

- The WESTINGHOUSE system, a conventional two-circuit and dual-circuit brake actuation system for tractor and trailer. After pressing the main control valve on the pedal, air flows from one reservoir to the rear brakes, from the other to the front brakes of the tractor, and at the same time, by actuating the relay valve, the trailer brakes begin to operate [28].
- The BOSCH system, a single-line pressure relief system used to actuate the braking mechanisms of the wheels of a motor vehicle and all the trailers towed by it. The system has compressed air reservoirs connected in series, built-in on each vehicle (tractor or trailer). The air pumped by the compressor is cleaned in a filter equipped with a connection for pumping the vehicle's tyres. A double control valve regulates the supply of compressed air to the vehicle's wheel brake actuators and actuates the control valve of the braking system of the first trailer [29].

In modern public transport vehicles, EBS—Electronic Braking System—can still be distinguished (Figure 5), which includes such elements as:

- ABS wheel anti-lock braking system,
- ASR drive wheel anti-slip system.

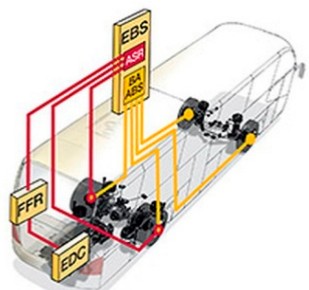

**Figure 5.** Diagram of the electronic braking system [27].

The use of electronic braking systems extends the life of many mechanical components of public transport vehicles.

The ABS system uses sensors to record the speed of each wheel during the braking process and adjusts the braking force separately for each wheel. In addition, it prevents the wheels from locking. Thus, even in the course of full braking on slippery surfaces, the vehicle maintains directional stability and steering.

ASR prevents the drive wheels from spinning in place when starting or accelerating owing to the EBS controller's recognition of EBS by means of wheel speed sensors. To adjust the ASR brakes, the EBS controller uses the same wheel speed sensors as for ABS, but only acts on the drive wheels, when only one wheel of the drive axle slips, as a result of braking the slipping wheel, the drive slip is reduced and the differential transmits the remaining drive torque to the other wheel with grip.

The advantages of using electronic braking systems are [27]:

- ASR drive wheel anti-slip system,
- braking with the maximum physically possible braking force with a shorter reaction time (shorter braking distance),
- controlled braking with high driving stability and also controllability in the case of emergency braking (braking on different surfaces),
- saving tyres (even wear on the circumference, no turning of wheels in place),
- better traction on slippery surfaces (i.e., ice, snow, gravel, or paving),
- increased driving safety (no rear-end fishtail).

### 3.3. Steering

The steering system is a component of a motor vehicle, which is responsible for safety and driving. It is intended to enable the vehicle to change direction smoothly and in accordance with the principles of car traffic mechanics, i.e., to set the steered wheels in such a way that the vehicle can move on a track of its own choice without lateral slip. The steering system should also ensure that the steered wheels are turned with as little force as possible applied to the steering wheel, that the steered wheels return to moving straight and that this direction of travel is maintained automatically [20,26,30].

The steering system consists of 3 parts [31]:

- Steering column,
- mechanical steering gear,
- return mechanism.

Figure 6 shows a diagram of the steering system and its components.

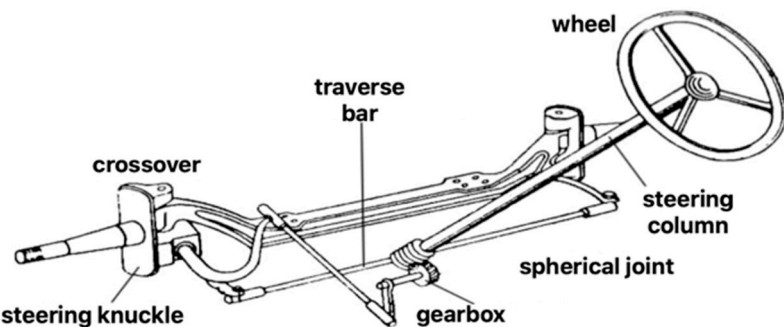

**Figure 6.** Diagram of the steering system [32].

The most frequent failures of the steering system of motor vehicles are excessive slack in its components, i.e., connection of steering rods and steering knuckles, steering column joints, as well as in the steering transmission. This is manifested by clunk or difficulties in maintaining the driving direction [3].

The cause of play may be natural wear of some elements. They also occur as a result of driving on bumpy and potholed roads. In order to reduce the wear and tear, fast or sudden invasion of obstacles, holes in the roadway and kerbs should be avoided. Another reason for slack is damaged rubber covers for ball joints, which do not protect their parts from contamination, causing premature wear or damage to the joint.

The technical condition of the steering system has a significant impact on active and passive safety of the vehicle. Therefore, during vehicle testing carried out at periodic inspections to check the vehicle's efficiency of operation, the steering system should be thoroughly checked by a diagnostician.

A solution improving the wear and tear of the steering system components is an intelligent steering system aimed at assisting the driver of the vehicle, including: Keeping the vehicle within the lane, automatically stopping the vehicle at bus stops, and detecting an unfavourable (affecting the system's lifetime) type of ground on which the vehicle is moving. In addition, it is recommended that drivers be trained to drive optimally in urban conditions, as well as to carry out ongoing checks for steering clearances in vehicles.

### 3.4. Bus Door Opening and Closing Mechanism

The door opening/closing mechanism (Figure 7) operating in buses is an important element of passenger transport in the city. It allows the customers of public transport companies to use services smoothly. However, its functioning depends on the number of vehicle stops at the stops where it is opened/closed, which often results in mechanical failures of the mechanism's components.

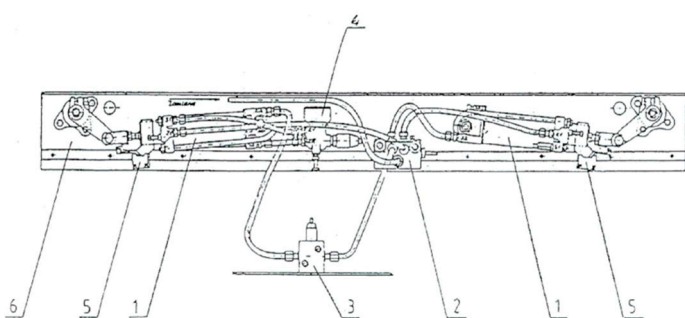

**Figure 7.** Mechanism of opening/closing the bus door: 1—actuator set with lever, 2—electropneumatic valve, 3—three-way connector, 4—valve switch, 5—limit switch, 6—mechanism basis [32].

Bus door opening and closing mechanisms are known in two basic variants [12]:

- With a door that opens inwards,
- with a door that opens outside the bus.

In the first solution, the door leaves move along a curve in the opening process, and in the final phase of movement they are arranged on the sides of the door opening perpendicularly to its plane, inside the vehicle. In the other solution, the door leaves are folded in the middle of the width and both parts of each leaf are arranged similarly to the first solution. The disadvantage of this solution is that the opening door occupies the area inside the vehicle, as well as the possibility of collision between the opening door and the passenger.

In the other solution, the door leaves pass outside the vehicle in a circular motion while opening, while maintaining the parallelism of the door leaf plane to the vehicle side. The disadvantage of this solution is that the door is significantly moved away from the vehicle body, which may lead to a collision with a passenger standing close to the vehicle, or clamping the passenger.

Pneumatic actuator 1 is the source of drive in the mechanism shown in Figure 7. The closing of the door is performed by the pushing side of the piston and the rotary-cam mechanism, which converts the piston's sliding motion into a rotating door column, simultaneously lifting to lock it in the final phase of the closing motion. The actuator also allows for reverse execution (opening the door in case there is too much resistance when closing) and emergency stopping of the door when opening it (after exceeding the constant resistance torque). Control is carried out by means of a pneumatic distributor (solenoid valve (2)), a reed contact proximity sensor with a light diode, a limit switch (5), a and relay. The proximity sensor acts as a reversal switch at the end of the door closing motion and the limit switch as a reversal indicator. The reverse (during closing) is executed in the following way: After encountering an obstacle, the door leaf stops with increasing resistance until the limit switch, which then sends a signal to open the door. In this case the door can be closed after the driver presses the closing button again.

One of the solutions to reduce the wear and tear of this part of the bus is patent number 380,237 of 18.07.2006, which aims to increase the safety of the bus door and improve the strength of the mechanism [33]. The device is based on a system of rotary-sliding doors and a drive mechanism with longitudinal and transverse guides.

### 3.5. Gearbox

The gearbox is the basic system, and one of the most important in motor vehicles. Its function is to transmit the torque generated by the engine to the vehicle's wheels so that the vehicle can move smoothly by providing the appropriate ratio.

At present, there are several types of gearboxes, based on different design solutions, but performing the same basic task, i.e., changing the ratio. The most common gearboxes used in the automotive industry are:

- Manual boxes,
- step-by-step automatic boxes,
- semi-automatic stepwise transmissions,
- stepless gearboxes.

In city vehicles, automatic transmissions are the most common types of transmission (Figure 8), as their main advantage is to relieve the driver from the need to change gears and press the clutch pedal, where it plays a key role in city driving.

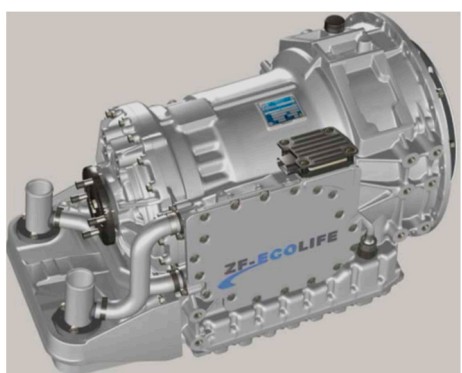

**Figure 8.** Automatic transmission for urban vehicles (ZF Eco life) [34].

As shown in Figure 8, automatic transmission is one of the most commonly used transmissions in public transport vehicles, capable of transmitting propulsion torque up to 2000 Nm. A vehicle with this type of gearbox consumes about 5% less fuel compared to the previous versions of the gearbox [30]. This is possible owing to the optimization of gear ratios and the introduction of a new torque converter. Vehicles equipped with this type of gearbox are characterised by better acceleration, although the gear change takes place in the lower speed range. Moreover, operation in the lower speed range also has an impact on lower noise emission, both inside and outside the vehicle [34].

In most cases, the operation of the gearbox is only connected with oil control and its refill, replacement, etc., if necessary. It is the most remarkable element of operation, as its durability and correct operation of automatic transmissions depend on it.

### 3.6. Leveling Electronics for the Vehicle

The last analysed element is the vehicle levelling electronics, as well as enabling the so-called "kneeling", which consists in lowering the right part of the vehicle by about 70 mm in such a way as to reduce the difference between the bus and the kerb at the bus stop, making it easier for people with disabilities or parents with prams to get inside. For this purpose, ECAS, based on Electronically Controlled Air Suspension, is most commonly used. ECAS (Figure 9) consists of several elements: Control unit (ECU), position sensors (sensors), and air pressure in the suspension bellows, electromagnetic valve unit, and remote control device [35].

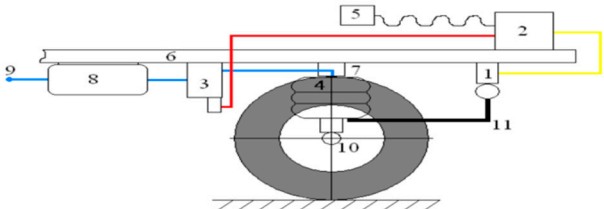

**Figure 9.** An example of Electronic Level Control (ECAS) for a pneumatic suspension system. 1—position sensor (sensor), 2—ECU (electronic control unit), 3—electromagnetic valve assembly, 4—pneumatic suspension bell, 5—remote control unit, 6—vehicle frame, 7—pressure sensor, 8—compressed air tank, 9—compressed air connection, 10—vehicle axle (front or rear), 11—tie rods [35].

The suspension is adjusted in the following way: A signal from the sensors is sent via electrical wires to the control unit (ECU), where after signal processing, the ECU performs a comparative analysis of the received signal with the reference signals stored in its RAM memory. If the ECU detects that the received signal differs from the programmed one, it works by sending an electrical signal to the solenoid valve unit, which orders to change the volume of compressed air in the diaphragm bellows. The signals are sent from the sensors to the ECU every 0.02 s [35].

The optimization action of this system is the installation of the device's structural elements, which will function faultlessly in variable climatic conditions, especially in low temperatures and high humidity. Moreover, the system is to have an effective dehumidification system and a system equipped with a heated automatic dehumidifier and a heated dehumidifier. All the connecting pipes should be rigid, and pneumatic fittings should be made of stainless materials.

## 4. Use of Decision Boards in City Vehicle Operations

Decision tables belong to the algorithmic information flow patterns. The presented block, economic and decision-making analysis, shows which elements are prioritised and which can be presented in the queue sense [36]. The decision model describes the selection of the type of wear part in city vehicles that wears out the fastest. Naturally, you can also perform fatigue and strength analysis of vehicle parts, which is in fact the standard method for finding vehicle parts that degrade in the fastest way possible. However, the analysis of fatigue and strength of parts will depend on the quality of the parts made and the manufacturer. Original parts recommended by the manufacturer are not always used in vehicles or transport devices. Frequently, spare parts are used that do not differ in quality from the original parts, and in terms of price. Therefore, this article focuses on the analysis of the operating parts of the vehicle using a decision model to pay attention to the parts that wear out the fastest.

Another test process may concern the analysis of fatigue and strength of individual parts of one type of the operating subsystem (taking into account different manufacturers, etc.). In view of the above, seven decision rules were proposed to describe the aforementioned elements of the vehicle:

- rule 1: Steering,
- rule 2: Gearboxes,
- rule 3: Clutch,
- rule 4: Levelling electronics for the bus,
- rule 5: Door control actuators,
- rule 6: Brake pads and discs,
- rule 7: Tyres.

Table 2 shows a decision model for the wear and tear of wear parts in urban vehicles. The rules contained in the table have been selected by means of the analysis carried out in Chapter 3 of this article, which determines whether particular parts constitute an input to the analysis. The action sets, on the other hand, are responses to the conditions.



**Table 2.** Decision-making model for the operational wear and tear of city vehicle mechanics [Own study].

| | | R1 | R2 | R3 | R4 | R5 | R6 | R7 |
|---|---|---|---|---|---|---|---|---|
| W1 | Are front brakes (brake pads and discs) an input? | - | - | N | - | - | Y | N |
| W2 | Are the rear brakes (drum and brake disc) an input? | - | - | N | - | - | Y | N |
| W3 | Is the master cylinder an input? | - | N | N | - | - | N | - |
| W4 | Is the brake pedal an input? | N | N | N | - | - | N | - |
| W5 | Are the cables connecting all the parts of the vehicle to each other? | Y | Y | Y | Y | Y | Y | N |
| W6 | Is the steering column an input? | Y | Y | N | N | - | N | - |
| W7 | Is the input element mechanical steering gear? | Y | N | N | Y | - | N | - |
| W8 | Is the input element a return mechanism? | Y | N | - | Y | - | - | N |
| W9 | Is there an inward opening door? | - | - | - | Y | Y | - | - |
| W10 | Is the door open to the outside? | - | - | - | Y | Y | - | - |
| W11 | Are manual boxes an entry point? | N | Y | Y | - | - | - | - |
| W12 | Is the input element automatic step-boxes? | N | Y | N | - | - | - | - |
| W13 | Are the input elements semi-automatic step boxes? | N | Y | N | - | - | - | - |
| W14 | Are there stepless boxes at the entrance? | N | Y | N | - | - | - | - |
| W15 | Is the vehicle's tyres an entry point? | N | - | - | - | - | N | Y |
| W16 | Is the pressure sensor an input element? | - | - | - | Y | Y | - | Y |
| C1 | Road conditions | X | - | X | - | X | X | X |
| C2 | Field conditions | X | X | X | - | - | X | X |
| C3 | Climatic conditions | - | X | X | X | X | X | X |
| C4 | Type of work to be done | X | - | - | X | X | - | - |
| C5 | Use of inappropriate lubricants | X | X | X | - | X | X | - |
| C6 | Use of inappropriate oils | - | X | - | - | - | X | - |
| C7 | Incorrect performance of technical inspections | - | X | X | - | - | X | X |
| C8 | Understatement of initial signs of wear | - | - | - | - | X | X | X |
| C9 | Non-compliance with manufacturer's recommendations | X | X | X | X | X | X | X |
| C10 | Wear of vehicle suspension components | - | - | - | - | - | X | X |
| C11 | Traveling comfort | - | X | X | X | X | X | X |
| C12 | Driving comfort | X | X | X | - | X | X | X |
| C13 | Number of accidents caused by a tyre failure | X | X | - | - | - | X | X |
| C14 | Braking and accelerating | - | - | - | - | - | X | - |
| C15 | Turning the wheels during manoeuvring | X | - | - | X | - | X | X |
| C16 | Hitting the tyre sides against the kerbs | X | - | - | - | - | X | X |
| C17 | Implementation of fully fledged vehicle courses | - | X | X | X | X | X | X |
| C18 | Tyre saving (even wear on the circumference, no turning of the wheels in place) | - | - | - | X | - | - | X |
| C19 | Better traction on slippery surfaces (i.e., ice, snow, gravel or paving) | X | - | X | X | - | X | X |

The presented structure of the decision board corresponds to the structure of the decision boards. The lines describe the conditions of wear and tear of the vehicle's operating parts and elements that directly affect them. The rules are described in columns R1 ÷ R7.

The result of the analysis was obtained by associating the logical values of meeting or failure to meet the conditions required for the analysis of the vehicle operating wear. It can be seen that the conditional and functional parts of these rules are multi-element.

Figure 10 shows an example of the ends of the analysed dendrite. First of all, attention should be paid to the possibilities offered by the use of decision tables. In some cases the terminations result from the collected initial data, which were presented in Chapter 3 of this paper.

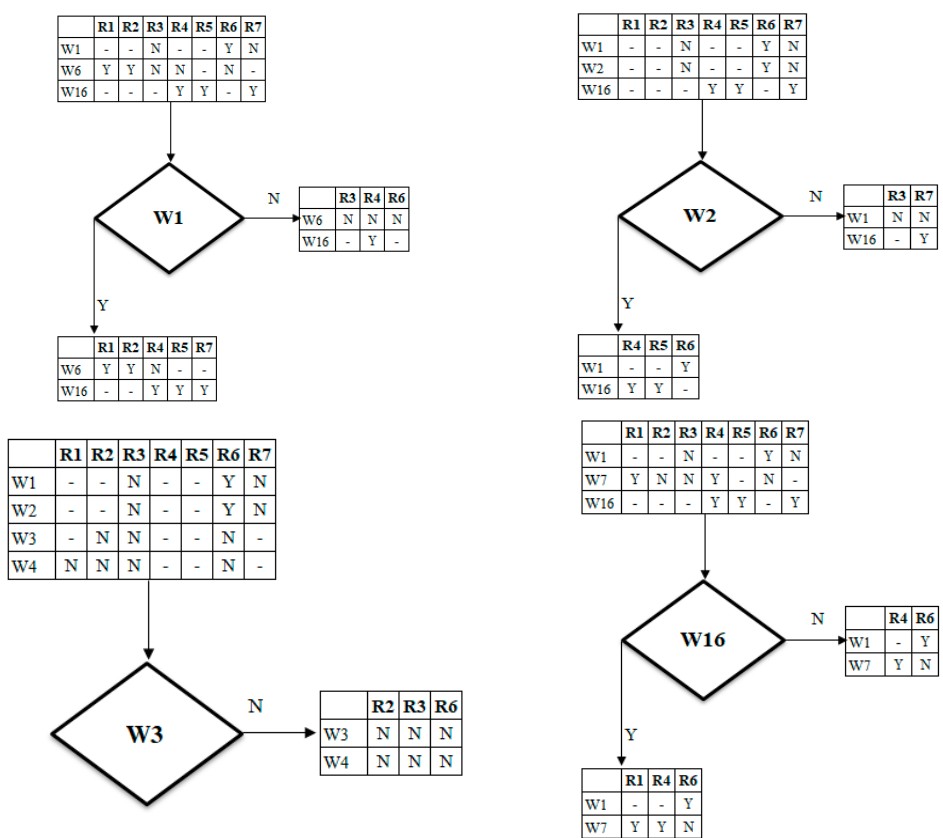

**Figure 10.** Examples of spanning tree endings from the decision board layout for W1–W16 [Own study].

The presentation of the numerical taxonomy for decision-making councils enables the introduction of stages with different levels of detail. The above issue corresponds to the appearance of the so-called partial needs, which are related to the implementation of the solution of the investigated problem. The hierarchical approach corresponds to the structure of the design process of passenger and freight transport in the city [37]. Various examples of possible endings of spanning tree are presented above. According to the decision analysis carried out, rule 1, as the least influencing the vehicle's wear and tear, was eliminated. The process of elimination involved rejecting those rules where the least correlation between the rule and the conditions appeared. On the other hand, the most worn elements are gearboxes, clutches, vehicle levelling electronics, and brake pads and discs. During the analysis, rules W3, W5, W8, W9, and W10 were also eliminated. Rule 1 was also eliminated due to the fact that during the decision analysis the smallest relation between this rule and the adopted conditions appeared. "The solution of the decision-making process in conditions of uncertainty, consisting in determining the optimal decision, can be presented on a graph, called a spanning tree, or—as this graph is assumed to be called in the theory of decision making—a decision tree" [38].

## 5. Conclusions

City vehicles are mainly used to transport large numbers of people and goods on relatively short routes. Taking this into account, these vehicles must have wide doors, the lowest possible floor, wide and comfortable gangways between the seats and many standing places, and they must provide passengers with a safe journey on this mode of transport. Therefore, mechanical components should be able to withstand long journeys in such difficult conditions as the city. Table 3 shows the measures proposed by the authors after consultation with vehicle drivers to reduce the wear and tear of vehicle components. These actions have a significant impact on the quality of travel, safety, and costs associated with the replacement of elements [39].

**Table 3.** Proposed actions limiting the operational wear of mechanical components of city vehicles [Own study].

| No. | Name of Vehicle Component | Proposed Actions |
|---|---|---|
| 1. | Tyres | • Limit the hitting tyres against the kerbs by carefully driving the bus to the stop,<br>• paying attention while driving to holes, protruding elements in the road infrastructure,<br>• a selection of tyres from renowned brands,<br>• mandatory daily inspection of tyres by drivers and technicians, |
| 2. | Braking system | • use of electronic braking systems,<br>• systematic brake pad and disc inspections,<br>• restriction of the use of the brakes when travelling to stops and traffic lights (engine braking),<br>• limit violent inhibitions, |
| 3. | Steering | • the use of intelligent steering,<br>• introducing driver training on optimal driving in urban conditions,<br>• limiting violent turning by rational driving,<br>• adjustment of the routes for curves of more than 90 degrees,<br>• performing ongoing management clearance checks, |
| 4. | Door opening and closing mechanism | • the use of modern mechanisms for opening and closing doors,<br>• paying attention when using the mechanism to the possibility of collision between the door and the passenger,<br>• not closing or opening the door at short intervals,<br>• current controls of the mechanism (slack, malfunctioning), |
| 5. | Gearbox | • gearbox oil control,<br>• systematic oil refilling (if necessary),<br>• oil change (if necessary),<br>• the use of electronic transmissions in vehicles, |
| 6. | Levelling electronics | • introduction of a system for dehumidifying the installation together with its heater,<br>• introduction of a system that heats up the oil separator,<br>• use of rigid connecting cables,<br>• use of stainless steel materials, pneumatic connectors,<br>• performing ongoing checks on the proper functioning of the electronics. |

In view of the analysis carried out, it can be concluded that the signs of wear and tear on the components of transport vehicles should not be underestimated, as the safety of the goods transported or the comfort of urban travellers in urban vehicles depends on them.

The common actions of the individual elements are their ongoing control in order to avoid the undesirable effects of wear and tear, as well as the training of those driving such vehicles to use them properly and economically.

The presentation of the numerical taxonomy for the decision boards enables the introduction of stages with different levels of detail. The above issue corresponds to the emergence of the so-called partial needs, which are connected with the implementation of the solution to the problem under investigation. The hierarchical approach corresponds to the structure of wear and tear of wear parts in the aspect of urban safety [37]. Decision tables make logical decisions a reality. The decision table only defines logical rules. Since the order of conditions is not given and the order in which the rules are defined is irrelevant, the conditions part of the table does not affect the implementation procedures. In this way, the decision table notation provides access to the tools of the implementation tool design. Under conditions deemed important, some options will be better than others. Ten choices have so far been recognised as the work area of the user and decision table builder, and generally followed up on this common sense when making decisions [40]. As a result, the user needs to have the results of the decision table compared to the flowchart that is meaningful to his organisation/company/situation. A decision board uses the available information and considers the various combinations to help one make a decision. In this way, the decision tables not only provide a clearer logic to follow during the process, but also provide a notation to enable transition to transition [41].

Various examples of possible endings of spanning tree are presented in the paper. According to the decision analysis carried out, Rule 1 was eliminated as the least significant factor influencing vehicle wear and tear and the least significant factor influencing urban safety. On the other hand, the most worn out elements are gearboxes, clutches, bus levelling electronics, and brake pads and discs. The decision-making model made it possible to identify the factors that have the greatest impact on reducing safety in urban space. The most important and most influential factors affecting the urban vehicle system include terrain conditions, climatic conditions, the use of improper lubricants, incorrect performance of technical inspections, non-compliance with the manufacturer's recommendations, comfort of driving, the implementation of fully fledged vehicles, and traction on slippery surfaces. These elements serve not only the safety of passengers and goods taking part in urban transport, but also the external environment [42]. An analysis of the decision-making model in terms of the operating systems of city vehicles is an extremely important link to ensure safety in urban areas. It should be taken into account that more and more metropolitan areas are preventing passenger cars from entering the city centre, leaving the possibility to travel only by public transport. Therefore, it is such an important factor to determine the impact of wear parts on urban space safety. Using the decision-making model, it is possible to adjust the factors influencing the examined wear parts in terms of specific decision-making conditions. During the model analysis, conditions W3, W5, W8, W9, and W10 were eliminated, as elements disturbing the results of decision analysis. In fact, it is accepted that the steering and braking systems are considered to be the most important from the point of view of safety. However, on the basis of the conducted tests, selected rules, conditions, and activities, the above-mentioned conditions have the least impact on the control system. It does not mean that the control system is redundant, but it can be influenced by other conditions that were not taken into account during the above analysis. Due to this fact, this rule was eliminated during the decision analysis.

Decision tables allow one to easily interact and collaborate with the analysis of data, which a decision should be based on. The decision board provides a simple and understandable interface that anyone can read and understand easily. Decision table users and constructors can both read, discuss, and work with the data contained in the decision table. It is a very good decision support tool. A clear tabular representation also prevents errors from constructors, system analysts, or programmers. Due to all these benefits, one can observe some aspects during the decision making process that were not taken into account in the original version, and may turn out to be key factors for the further operation of machines, devices, organizations, companies, etc. [43].

The aim of this article was to draw attention to the management of city transport from the perspective of the elements included in the city vehicle. The conducted analysis is

aimed at drawing attention to those elements that make up the city vehicle, which are the most wear-and-tear susceptible, and therefore require more frequent control by operators providing public transport services. Another element of the research can be the specific parts of each of the components of the city vehicle described above in terms of strength and fatigue, e.g., by comparing the same parts from different manufacturers, at different times of the year, etc. Based on the data, a decision analysis was carried out to support decisions about the elements that wear out. The above does not mean that the communication operator should not check the technical condition of the entire vehicle in accordance with the law and safety rules [41].

The test carried out has shown how important it is to wear out structural elements of motor vehicles. However, at this point, it should be noted that a very important and developmental factor influencing urban transport is the introduction of electric vans travelling around the city. It is not so much by lowering the environmental impact factor as by using modern vehicle parts that the safety of urban road users can be improved.

Research will continue to determine the impact of the use of electric vehicles in urban space, which should be based primarily on the use of electronics to increase safety when transporting goods in urban space.

**Author Contributions:** Conceptualization, D.M.; methodology, D.M. and M.D.-G.; validation, D.M. and E.K.; formal analysis, E.K.; investigation, D.M. and M.D.-G.; resources, M.D.-G.; writing—original draft preparation, D.M. and M.D.-G.; writing—review and editing, D.M. and E.K.; visualization, D.M.; supervision, D.M. and E.K.; project administration, D.M.; funding acquisition, E.K. All authors have read and agreed to the published version of the manuscript.

**Funding:** This research was funded by National Science Centre (NCN) grant number UMO-2012/05/B/HS4/04139.

**Institutional Review Board Statement:** Not applicable.

**Informed Consent Statement:** Not applicable.

**Data Availability Statement:** No new data were created or analysed in this study. Data sharing is not applicable to this article.

**Conflicts of Interest:** The authors declare no conflict of interest.

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
