# Peer review of "A Decision-Making Model on the Impact of Vehicle Use on Urban Safety"

_sustainability, doi:10.3390/su13063585_

Round 1
Reviewer 1 Report
The topic of article is interesting and important; therefore, the publication of the paper is recommended. However, it is suggested to correct, rewrite, clarify or rework some part of text.
In general, the first part of the article is well-described, containing many important information about the topic. Yet, the aim of the paper (Line 68-68) should be more detailed and clearer. Nevertheless, the section “Materials and Methods” does not present the used methods of the article.
The most essential and significant part of the paper is the presented decision board. However, the description of it is too brief. A more explicit and comprehensive explanation of the usage of the board and the elimination process would be reasonable for the better understanding, as well as to emphasize the findings of the paper.
Other suggestions:
Line 80-81 – The paper should explain which parts were included and which are not.
Figure 1. – The names of the elements are missing (i.e. Zd, Zgr, Td, Te’, Te etc.)
Figure 2. quality shall be better
L 111 Since the referred Figure 3. presents buses, “Tyres are one of the most important parts in city vehicles, especially in vans and buses” would be better.
L 114 Figure 3 - (b) should be placed under the right photograph
L 145 “They are brake pads that are pressed against the pads” - clarification required
L 175 - kg/cm2
Figure 5 is a bit too small
L 215 instead of period --> comma
L 287 At first it is not clear that the numbers are referring to Figure 7. A better way would be:
(solenoid valve (2)), reed contact proximity sensor with a light diode, limit switch (5) and relay
L 374-375 says “On the other hand, the factors are described by the symbols X and '-'.” but the table marks it as an empty cell. Rephrasing is recommended.
L 415 “failure to meetthe conditions required” a space between ’meet’ and ’the’ is missing
Table 2 and Table 3 is basically the same. Combining them would be reasonable. (E.g. renaming the first coulomb from 1, 2, 3 … to W1, W2, W3 …)
Figure 10. requires more clarification in the text body.
L 426-432 contains one of the most important findings of the paper. Therefore, it requires deeper discussion. How the elimination process took place?
Table 4. – The font size of the table’s name is too small and not bold. The spacing of the bullet points vary. In the point 3. Steering – “Introducing driver training on optimal driving in urban conditions,” should start with lowercase letter (like the other points).
Author Response
Hello,
Thank you very much for pointing out your very valuable comments on the article. We have tried to take into account all the comments suggested by the reviewer. The detailed description of the individual remarks was as follows:
- the aim of the paper has been changed and clarified to reflect the essence of the article. In addition, the description regarding the methodology of the paper has been clarified to include a description of the decision tables,
- the method of implementation and use of decision tables and the process of elimination of particular conditions has been extended in the new version of the article,
- the description in lines 80-81 contains the distribution of one part in the vehicle. This is a generalized model of the wear curve of vehicle parts,
- the graph in Fig. 10 shows the curve of wear of vehicle parts taking into account the OPR'B curve. The abbreviations on the axes of the graph represent the service life of the respective components respectively,
- the quality of Figure 2 has been improved,
- the sentence in line 111 has been corrected according to the reviewer's remark,
- point 'b' in figure 3 has been moved to the correct place,
- the sentence in line 145 has been framed and clarified,
- the unit kG/cm2 referring to pressure has been replaced by a figure expressed in Pascals for better understanding,
- figure 5 has been enlarged as noted,
- on line 215, the full stop was replaced by a comma
- on line 287 the text has been changed as suggested by the Reviewer,
- the information in lines 374-375 has been aligned with the table. The table has been amended,
- in line 415 has been amended in line with the reviewer's comment,
- table 2 and 3 have been reduced to one table in accordance with the Reviewer's note,
- additional commentary under the figure was made in the text. Other
- information concerning the figure was included as general conclusions in the last chapter,
- the paper contains information on how the process of elimination of individual conditions took place, general conclusions are included in the final part of the paper,
- the technical part of the table has been modified.
Finally, I would like to thank you very much for your comments on the paper. They are very valuable to us and will certainly improve our work. I also hope that the corrected parts of the paper will fully satisfy the reviewer. The corrected work has been added to the appendix.

Reviewer 2 Report
Lines 107-109 "The article will analyse the above-mentioned parts used in vehicles moving around in urban space, with particular emphasis on the proposed measures to reduce the process of wear and tear of elements [18]." Why there is a reference in the end of the sentence describing the content of the paper?
The title of the section 2 relates to materials and methods. There are no methods described in this chapter. Instead there is a large description of parts and elements of buses (points 2.2-2.6) which is not relevant to the decision making process presented in the paper. there is no need to describe the detailed structure and diagram of the braking system or steering system.
Figure 10 presents on two pages spanning tree ' endings from the decision board layout for W1-W16. It doesn't give any additional information to the paper - one spanning tree as an example would be enough.
In section 3 rules W1-16 are not explained which reduces the added value of the presented method.
The conclusions are not supported by the presented research i.e. proposed actions limiting the operational wear of mechanical components of city vehicles do not result from the method presented in section 3 of the paper.
21 out of 34 (62%) of all references cited in the paper are available only in Polish. As Sustainability is a journal aimed at international readers, cited references should be available in English. The reference list must be changed before further processing of the paper.
Author Response
Hello,
thank you very much for your review. All the comments made have been taken into account and corrected in the work. As far as the individual comments are concerned:
- in lines 107-109 an appendix was indeed included by mistake. It has been moved to the correct place.
- the structure of the work has been changed. Chapter 2 has been renamed Chapter 3. On the other hand, the chapter relating to the description of the decision tables was included as a description of the research tools included in the paper. Obviously the description of the individual vehicle parts forms the basis for the development of the decision table.
- figure 10 has been modified and in this version it only contains examples of solutions to the developed decision tree.
- the paper states that the rules were developed on the basis of an analysis of urban vehicle parts,
- proposed mitigation measures have been included in the text in order to provide the reader with ways to respond to the wear and tear of urban vehicle components. Drivers of urban vehicles were consulted on these proposals.
- The literature in the paper has been modified and supplemented by English-language items. Unfortunately, the remaining Polish items could not be changed as in some cases they constitute the foundation of the given part of the article.
I very much hope that the amendments made will fully satisfy the honourable reviewer. A revised article has been added as an attachment.

Reviewer 3 Report
The article is very interesting. The issues raised in it are very important from the point of view of sustainable development and transport. The aim of the article is to show actions to reduce the operating wear and tear of city vehicles as well as some of the vehicles running in the city on the example of city buses. The goal was formulated correctly and was achieved.
The study lacks a typical literature review, but in my opinion, this form is also consistent and logical.
Very interesting graphic material also deserves attention.
Please note that the summary is too long. According to the publisher's guidelines, it should not exceed 200 words. Please shorten if possible.
Author Response
Thank you very much for your review. After reviewing the comments, the paper has been modified. I am sending the revised paper as an attachment. As for the abstract, it was shortened to 200 words.

Round 2
Reviewer 2 Report
The quality and structure of the papaer has improved. I still think that the citations should reffer to bibliography in English but you the Authors have slightly improved this area as well.